# A New Theoretical Approach of Wall Transpiration in the Cavity Flow of the Ferrofluids

**DOI:** 10.3390/mi10060373

**Published:** 2019-06-04

**Authors:** Abuzar Abid Siddiqui, Mustafa Turkyilmazoglu

**Affiliations:** 1Department of Basic Sciences, Bahauddin Zakariya University, Multan, Punjab 60000, Pakistan; 2Department of Mathematics, Hacettepe University, Ankara 312, Turkey; turkyilm@hacettepe.edu.tr

**Keywords:** ferrofluids, Kelvin force, Lorentz force, lid-driven cavity flow, permeability

## Abstract

An idea of permeable (suction/injection) chamber is proposed in the current work to control the secondary vortices appearing in the well-known lid-driven cavity flow by means of the water based ferrofluids. The Rosensweig model is conveniently adopted for the mathematical analysis of the physical problem. The governing equation of model is first transformed into the vorticity transport equation. A special finite difference method in association with the successive over-relaxation method (SOR) is then employed to numerically simulate the flow behavior. The effects of intensity of magnetic source (controlled by the Stuart number), aspect ratio of the cavity, rate of permeability (i.e., $\alpha_{p}=\frac{V_{0}}{U}$), ratio of speed of suction/injection $V_{0}$ to the sliding-speed U of the upper wall of a cavity, and Reynolds number on the ferrofluid in the cavity are fully examined. It is found that the secondary vortices residing on the lower wall of the cavity are dissolved by the implementation of the suction/injection chamber. Their character is dependent on the rate of permeability. The intensity of magnetic source affects the system in such a way to alter the flow and to transport the fluid away from the magnetic source location. It also reduces the loading effects on the walls of the cavity. If the depth of cavity (or the aspect ratio) is increased, the secondary vortices join together to form a single secondary vortex. The number of secondary vortices is shown to increase if the Reynolds number is increased for both the clear fluid as well as the ferrofluids. The suction and injection create resistance in settlement of solid ferroparticles on the bottom. The results obtained are validated with the existing data in the literature and satisfactory agreement is observed. The presented problem may find applications in biomedical, pharmaceutical, and engineering industries.

## 1. Introduction

Ferrofluids are fluids that contain magnetic nanoparticles in suspension with a particle diameter in the range of 5 nm or less [1,2]. The magnetic particles are in a single-domain state with a corresponding intrinsic spontaneous magnetization and are typically made of magnetite or maghemite. These particles are thoroughly coated with a surfactant to inhibit clustering. They can exhibit translational and rotational Brownian motion [1]. The fine magnetic particles are derived by the external magnetic source that causes the ambient motion of the carrier fluid [2]. Dynamics of the ferrofluids—ferrohydrodynamics (FHD)—differ from the magnetohydrodynamics (MHD) in the manner that FHD obeys the Lorentz as well as the Kelvin forces while MHD supports the Lorentz force merely [1]. Ferrofluids are unique by their distinguished characteristics because there are no known naturally occurring magnetic liquids. Further, they are in distinction from MHD, the flow phenomena occur without the need for electrical current [3].

These magnetically responsive fluids are used in the industries remarkably. Initially the ferrofluid was invented by Papell [3] in 1963 and was utilized in NASA as the rocket propellant for use in a zero-gravity environment. Papell also claimed in his patent that “the said propellant may be oriented and attracted by the imposition of a magnetic field” [4].

One year later, Neuringer and Rosensweig [5] witnessed the idea of [4]. Besides this novel application of ferrofluids, they have tremendous use in engineering, in biomedical and pharmaceutical industries. For example, they are useful in viva imagining, hydraulic suspension pistons (with the intensity of the magnetic field permitting the suspension to be soft or hard as per requirement), constituting liquid seals for the drive shaft of hard disks, and dealing with the stepper motor dampers [6]. The ferrofluids have great value in manufacturing the loudspeaker [7]. They are also significant in pharmaceutical industry in order to transport the targeted drug delivery [8,9]. The incredible applications of ferrofluids in the biomedical industry for magnetic resonance imaging (MRI), magnetic separation, hyperthermia treatment, cell sorter, hyperthermia, and biosensors [10] should not be forgotten. Furthermore, the use of ferrofluids in microsystem technology is important since they can be incorporated into modern products like micro actuators [11,12] and ferrofluid-based microchip pumps and valves [13]. The ferrofluid particles are produced by precipitating the particles chemically with essential chemical reaction technique as mentioned in the articles [14,15]:


$${2F}_{e}^{3+}+F_{e}^{2+}+{8\mathrm{OH}}^{-}=F_{e}{\mathrm{OF}}_{e}{}_{2}O_{3}+{4H}_{2}O.$$


As said above, the story of ferrofluid was initiated by Papell [4]. Later on Neuringer and Rosensweig [5] developed its mathematical model regarding the motion of ferrofluids. They also proposed the name of this field as “ferrohydrodynamics”. Afterwards, Shliomis presented another model in 1968. However, Rosensweig model gained more popularity owing to its feasibility to apply. This model is also adopted in the present work. A lot of work has been done on this topic till to date theoretically as well as experimentally. For example, Patel et al. [16] used the ferrofluid for lubrication analysis in order for magnetic squeezing film in rough curved annular plates with assorted porous structures. The thermal effects on the ferrofluid are captured inside a right angled elbow channel in [17], which concluded that the Nusselt number (ratio of convective to conductive thermal transfer) increases with the increase in the intensity of magnetic field. Katsikis et al. [18] addressed a microfluidic platform for magnetic manipulation of water droplets immersed in bulk oil-based ferrofluid. Katsikis et al. [18] also drew attention on the future applications due to the potential biocompatibility of the droplets. The thermal and magnetic effects in T-shaped channel were studied in [19]. The mean thermal transfer of ferrofluid was claimed to rise more than 64% when the magnetic field is applied. Kúdelčíka et al. [20] presented the thermal effects on the magneto-dielectrics behavior of oil-based ferrofluids by dielectric spectroscopy. The natural convection for a ferrofluid in a trapezoid cavity in the presence of an inclined magnetic field was examined in [21]. An unstable behavior of heat and fluid flow was shown when the magnetic field is orthogonal. Theis-Bröhl et al. [22] investigated self–assembly of monodisperse colloidal magnetite nanoparticles from a water-based ferrofluid onto a silicon surface. Sheikholeslami et al. [23] studied the ferro-nanofluid in a double-sided lid-driven cavity with a wavy wall in the presence of a variable magnetic field. Sheikholeslami et al. [24] simulated the thermal and magnetic influences on the ferrofluid flow in a semi-annulus enclosure. Tzirtzilakis et al. [25] treated the ferrofluids as the biofluids and examined the lid-driven cavity flow. Amirat et al. [26] made the comments on the solution of Shliomis model of ferrohydrodynamics. Lin [27] discussed and derived ferrofluid lubrication equation of cylindrical squeeze films with convective fluid inertia forces and application to circular disks. Tzirtzilakis [28] described the blood flow by adopting the ferrohydrodynamics model. Strek [29,30] investigated the thermal influences on the ferrofluids in the channel. Recently, a couple of interesting attempts were made to analyze the convection by ferrofluids in the cavity in Geridönmez [31,32]. In addition to this, Sheikholeslami et al. [33] investigated the thermal effects of the ferrofluid in the wavy wall cavity. In short, there is an extensive review on advances in the field of ferrohydrodynamics technology; one such can be referred to Raj et al. [34].

The well-known classical lid-driven cavity flow is important techno-scientifically because it is useful in processing foods and polymers as well as in topological mixing of liquids in the mixture [35,36,37]. The mixing can be produced by moving/sliding the wall of cavity rather than applying the internal rotors for this purpose. It has also significant applications in producing high-grade paper and photographic film [38]. A lot of work has been fulfilled on this flow problem in different aspects and fashions. Some of selected papers are [39,40,41,42,43,44,45,46,47]. Reinholtz [48] summarized very well the extensive review on this topic, which may be consulted by the interested reader.

Unlike the thorough review background aforementioned, whose common thread was to approach the problem from a traditional fashion in the present attempt, it is aimed to explore the steady flow of ferrofluid in a lid driven cavity. A variable and strong magnet is placed in the vicinity of the cavity. Additionally, the lower wall of the cavity is furnished with the suction/injection chamber as shown in Figure 1. Making use of such a chamber has not been attempted earlier, to the best of knowledge of the authors. This is completely a new idea in the fluid dynamics in which half wall is injecting and half is suctioning in order to preserve the law of conservation of mass. By virtue of this kind of permeability, it can be controlled the suspended solid ferroparticles to reside/settle down to the bottom. As a consequence, they remain suspended and obey Brownian motion for all the time. Moreover, the corner secondary vortices, which create the singular points or poles, can be shifted/decayed/vanished through this chamber.

The hydrodynamic flow problem is computationally studied by adopting the special finite difference method along with the successive over relaxation. The outcomes are simulated in the graphical form in terms of streamlines, equivorticity lines, and velocity plots, as well as in tabular forms comprising the optimal values and their locations of the stream function and the vorticity in the cavity.

The composition of this article is such that the mathematical formulation and numerical methods are presented in Section 2. The graphical and tabular results and discussions on them are given in Section 3. Finally, the main findings are noted in Section 4.

## 2. Mathematical Modeling and Analysis

We consider the steady flow of an electrically conducting and a magnetically responsive ferrofluid (e.g., ferrite nanoparticles dispersed and suspended in water) within a rectangular cavity (with dimensions a×b×c), whose cross-sectional view is shown in Figure 1. 

The flow is generated by sliding the upper wall (lid) of the cavity with velocity U from left to right. This flow setup is known as ‘lid-driven cavity flow’ in the literature [38]. We make an amendment in the lid-driven cavity flow by incorporating an injection/suction chamber adjacent to the lower wall of the cavity. This chamber has ability to make suction in the interval $0\leq x^{\prime }\leq a/2$ and injection in the interval $a/2<x^{\prime }\leq a$ of the cavity, provided that the rate of suction is equivalent to the rate of injection $V_{0}$. Please note that we have fixed here bifurcating point as $x^{\prime }=a/2$; it may vary though, according to the practical needs. A magnetic wire source is placed in the vicinity of the right wall of the cavity, parallel to the z− axis. The wire is passing through xy− plane at the point P(α, β, δ)=(a+0.001a,b/2,0). The magnetic source has variable intensity $H^{\prime }=(H_{1}^{\prime },H_{2}^{\prime })$, which is explicitly defined as [23]
(1)$$\;\begin{array}{l} H_{1}^{\prime }=\frac{\gamma_{m}}{2\pi }\frac{\left(y^{\prime }-\beta \right)}{\sqrt{{\left(x^{\prime }-\alpha \right)}^{2}+{\left(y^{\prime }-\beta \right)}^{2}}};H_{2}^{\prime }=\frac{\gamma_{m}}{2\pi }\frac{\left(x^{\prime }-\alpha \right)}{\sqrt{{\left(x^{\prime }-\alpha \right)}^{2}+{\left(y^{\prime }-\beta \right)}^{2}}};\\ \mathrm{where}\;H^{\prime }=\sqrt{{H_{1}^{\prime }}^{2}+{H_{2}^{\prime }}^{2}}=\frac{\gamma_{m}}{2\pi }\frac{1}{\sqrt{{\left(x^{\prime }-\alpha \right)}^{2}+{\left(y^{\prime }-\beta \right)}^{2}}}, \end{array}\}$$
whereas $\gamma_{m}$ is the magnetic field strength at the source P. In addition, the dimensionless magnetic field intensity $H=H^{\prime }/h_{0}$, where $h_{0}=\frac{\gamma_{m}}{2\pi a}$, is also simulated in Figure 2 for better understanding. 

We are interested in studying the aforementioned flow problem computationally. For this purpose, the best suitable model will be the modified Navier–Stokes equations for the ferrofluids (usually named as Rosensweig model) [5,31,32,33]. It can be described for the steady flow in the vorticity-velocity form in terms of the dimensional variables as
(2)$$\nabla^{\prime }\times \left[F_{K}^{\prime }+F_{L}^{\prime }+V^{\prime }\times \omega^{\prime }-\upsilon \nabla^{\prime }\times \omega^{\prime }\right]=0$$
where υ, $V^{\prime }$, and $\omega^{\prime }$ are, respectively, the kinematic viscosity, velocity and vorticity of the fluid such that the prime signifies for dimensional quantities. In addition, $F_{K}^{\prime }\left(=\mu_{0}\rho^{-1}\left[M^{\prime }\cdot \nabla^{\prime }\right]H^{\prime }\right)$ and $F_{L}^{\prime }\left(=\rho^{-1}\left[J^{\prime }\times B^{\prime }\right]\right)$ [28] represent the Kelvin and Lorentz forces per unit volume, respectively. Here, $B^{\prime }$, $H^{\prime }$, $M^{\prime }$, $\mu_{0}$, ρ and $J^{\prime }$ represent the magnetic flux density, the magnetic field intensity, the magnetization, the magnetic permeability of the vacuum, the fluid density, and the current density, respectively. Furthermore, if the relation between the current density and the magnetic field intensity viz., $\nabla^{\prime }\times H^{\prime }=J^{\prime }$ [28], and a well-known vector identity $\left(H^{\prime }.\nabla^{\prime }\right)H^{\prime }=0.5\nabla^{\prime }\left(H^{\prime }\cdot H^{\prime }\right)-H^{\prime }\times \left(\nabla^{\prime }\times H^{\prime }\right)$ [25] are used, then the aggregate of the magnetic forces yields as
(3)$$F_{K}^{\prime }+F_{L}^{\prime }=\left\{\mu_{0}\rho^{-1}\left(M^{\prime }\nabla^{\prime }H^{\prime }\right)+\chi \mu_{0}\rho^{-1}\left(J^{\prime }\times H^{\prime }\right)\right\}+\rho^{-1}\left(J^{\prime }\times B^{\prime }\right)$$

On the introduction of the relation between magnetization and magnetic field intensity, i.e., $M^{\prime }=\chi H^{\prime }$ [28] in Equation (3), we get the following ‘curl of the sum of the magnetic forces’, after a little bit simplification:(4)$$\nabla^{\prime }\times \left(F_{K}^{\prime }+F_{L}^{\prime }\right)=\mu_{0}\rho^{-1}\left(1+2\chi \right)\nabla^{\prime }\times \left(J^{\prime }\times H^{\prime }\right).$$

Here $J^{\prime }=\sigma \mu_{0}\left(1+\chi \right)\left(V^{\prime }\times H^{\prime }\right).$ Now, let us make Equation (2) dimensionless by introducing the following relations.
(5)$$\left\{V^{\prime },\;{x^{\prime }}_{i},\;\omega^{\prime },\;J^{\prime }\right\}=\left\{UV,{\;a}^{-1}x_{i},{\;\mathrm{Ua}}^{-1}\omega ,\;h_{0}U(1+\chi )\mu_{0}\sigma J\right\}$$
where, $\chi =\frac{M}{H}$ is the susceptibility and σ is the electric conductivity.

Accordingly, Equation (2), after using Equation (4), in dimensionless variables will take the form
(6)$$R\;S_{t}\;\nabla \times \left[\left(V\times H\right)\times H\right]-\nabla \times \left(\nabla \times \omega \right)+R\nabla \times \left(V\times \omega \right)=0$$
where $R\left(=\frac{Ua}{\upsilon }\right)$ is the Reynolds number and $S_{t}\left(=M_{n}K_{1}\right)$ is the modified Stuart number (the ratio of electromagnetic to the inertial forces). Moreover, $M_{n}\left(=\frac{\chi \left(1+\chi \right)\left(1+2\chi \right)h_{0}^{2}\mu_{0}}{U^{2}\rho }\right)$ is the magnetic number while $K_{1}=\frac{Ua\mu_{0}\sigma }{\chi }$ may be permoconductivity number [25].

If we assume a two dimensional flow, then Equation (6) in the stream function ψ and nonzero component vorticity E form in xy-plane will yield
(7)$$\nabla^{2}E+R\left[\frac{\partial \psi }{\partial x}\frac{\partial E}{\partial y}-\frac{\partial \psi }{\partial y}\frac{\partial E}{\partial x}\right]+RS_{t}\left[p\frac{\partial^{2}\psi }{\partial x^{2}}+q\frac{\partial^{2}\psi }{\partial y^{2}}+r\frac{\partial^{2}\psi }{\partial x\partial y}+s\frac{\partial \psi }{\partial x}+t\frac{\partial \psi }{\partial y}\right]=0,$$
where,
(8)$$E=-\nabla^{2}\psi$$

Here, the appearing quantities are
(9)$$\left\{p,\;q,\;r,\;s,\;t\right\}=\left\{H_{1}^{2},\;H_{2}^{2},\;2H_{1}H_{2},\;\frac{\partial \left(H_{1}H_{2}\right)}{\partial y}+2H_{1}\frac{\partial H_{1}}{\partial x},\;\frac{\partial \left(H_{1}H_{2}\right)}{\partial x}+2H_{2}\frac{\partial H_{2}}{\partial y}\right\}.$$

The boundary conditions can be summarized as
(10)$$\frac{\partial \psi }{\partial y}=0,\frac{\partial \psi }{\partial x}=-\alpha_{p}\;\mathrm{at}\;\mathrm{the}\;\mathrm{bottom}\;\mathrm{of}\;\mathrm{the}\;\mathrm{cavity}$$
(11)$$\frac{\partial \psi }{\partial y}=1,\frac{\partial \psi }{\partial x}=0\;\mathrm{at}\;\mathrm{the}\;\mathrm{lid}\;\mathrm{of}\;\mathrm{the}\;\mathrm{cavity}$$
(12)$$\frac{\partial \psi }{\partial y}=0,\frac{\partial \psi }{\partial x}=0\;\mathrm{at}\;\mathrm{left}\;\mathrm{and}\;\mathrm{right}\;\mathrm{walls}\;\mathrm{of}\;\mathrm{the}\;\mathrm{cavity}$$

Note that the boundary condition involves the permeability number $\alpha_{p}=\frac{V_{0}}{U}$. The boundary conditions on E will be described later on.

The above boundary value problem (BVP) comprising (7)–(9) cannot be solved analytically. Therefore, it is solved by the special finite difference method [49,50] and is described briefly at the mesh-points $\left(x_{0},\;y_{0}\right)$,$\left(x_{0}+\Delta x,\;y_{0}\right)$, $\left(x_{0},\;y_{0}+\Delta y\right)$, $\left(x_{0}-\Delta x,\;y_{0}\right)$, $\left(x_{0},\;y_{0}-\Delta y\right)$, $\left(x_{0}+\Delta x,\;y_{0}+\Delta y\right)$ and $\left(x_{0}-\Delta x,\;y_{0}+\Delta y\right)$ by the subscripts 0, 1, 2, 3, 4, 12  and 32 respectively. Here Δx and Δy represent the size of the mesh along x− and y− directions, respectively.

Let us split Equation (7) into three following equations as
(13)$$\frac{\partial^{2}E}{\partial x^{2}}+S\frac{\partial E}{\partial x}=\Theta (x,\;y),$$
(14)$$\frac{\partial^{2}E}{\partial y^{2}}+T\frac{\partial E}{\partial y}=-0.5\Theta (x,\;y)$$
(15)$$RS_{t}\left[p\frac{\partial^{2}\psi }{\partial x^{2}}+q\frac{\partial^{2}\psi }{\partial y^{2}}+r\frac{\partial^{2}\psi }{\partial x\partial y}+s\frac{\partial \psi }{\partial x}+t\frac{\partial \psi }{\partial y}\right]=-0.5\Theta (x,\;y)$$

Here, $\left\{S,\;T\right\}=\left\{-R\frac{\partial \psi }{\partial y},\;R\frac{\partial \psi }{\partial x}\right\}.$

Next, Equation (11) is approximated along grid line $y=y_{0}$
$\forall \;x\in \left[x_{0}-\Delta x,\;x_{0}+\Delta x\right]$ by applying the following transformation.
(16)$$\begin{array}{l} E(x,y)=L(x,y)e^{-P(x)},\\ P(x)=\frac{1}{2}\int_{x_{0}}^{x}S(\theta ,y)d\theta . \end{array}\}$$

On inserting Equation (14) into Equation (9) and applying the central difference approximation to the derivatives, we get
(17)$$-2L_{0}+L_{1}+L_{3}-\frac{{\left(\Delta x\right)}^{2}}{4}L_{0}\left[S_{0}^{2}+2{\left(\frac{\partial S}{\partial x}\right)}_{0}\right]={\left(\Delta x\right)}^{2}\Theta_{0}$$

Similarly, if Equation (12) is approximated along the grid line $x=x_{0}$
$\forall \;y\in \left[y_{0}-\Delta y,\;y_{0}+\Delta y\right]$ by setting the following local transformation.

where
(18)$$\begin{array}{l} E(x,y)=M(x,y)e^{-Q(x)},\\ Q(x)=\frac{1}{2}\int_{y_{0}}^{y}T(x,\theta )d\theta . \end{array}\}$$

On using Equation (16) in Equation (12) and applying the central difference approximation to the derivatives, we obtain
(19)$${\left(\frac{\Delta x}{\Delta y}\right)}^{2}\left[-2M_{0}+M_{2}+M_{4}\right]-\frac{{\left(\Delta x\right)}^{2}}{4}M_{0}\left[T_{0}^{2}+2{\left(\frac{\partial T}{\partial y}\right)}_{0}\right]=-0.5{\left(\Delta x\right)}^{2}\Theta_{0}$$

Now, multiply Equation (13) by ${\left(\Delta x\right)}^{2}$ and add it to Equations (15) and (17) after approximating the derivatives by the central difference method, we get
(20)$$L_{1}+L_{3}+{\left(\frac{\Delta x}{\Delta y}\right)}^{2}\left[M_{2}+M_{4}\right]-\frac{{\left(\Delta x\right)}^{2}E_{0}}{4}\left[S_{0}^{2}+T_{0}^{2}\right]-2E_{0}\left[1+{\left(\frac{\Delta x}{\Delta y}\right)}^{2}\right]+\Psi_{0}=0$$
where
(21)$$\Psi_{0}=RS_{t}\left\{\begin{array}{l} \left[p_{0}-\frac{r_{0}\Delta x}{2\Delta y}+\frac{s_{0}\Delta x}{2}\right]\psi_{1}+\left[\frac{q_{0}\Delta x}{{\left(\Delta y\right)}^{2}}+\frac{t_{0}{\left(\Delta x\right)}^{2}}{2\Delta y}\right]\psi_{2}+\left[p_{0}+\frac{r_{0}\Delta x}{2\Delta y}-\frac{s_{0}\Delta x}{2}\right]\psi_{3}+\\ \left[\frac{q_{0}\Delta x}{{\left(\Delta y\right)}^{2}}-\frac{t_{0}{\left(\Delta x\right)}^{2}}{2\Delta y}\right]\psi_{4}+\frac{r_{0}\Delta x}{2\Delta y}\left[\psi_{12}-\psi_{32}\right]-2\left[p_{0}+\frac{q_{0}\Delta x}{{\left(\Delta y\right)}^{2}}\right]\psi_{0} \end{array}\right\}$$

Let us write Equations (14) and (16) in the following fashion,
(22)$$L_{i}=E_{i}e^{P_{i}}\;\mathrm{and}\;M_{j}=E_{j}e^{Q_{j}};$$
where i=1, 3 and j=2, 4.

Next, let us expand the exponential in the power of their arguments, keeping truncation error $Ο\left[{\left(\Delta x\right)}^{4}\right]\Lambda Ο\left[{\left(\Delta x\right)}^{2}{\left(\Delta y\right)}^{2}\right]$ and put the resulting expressions in Equation (20), we obtain
(23)$$\begin{array}{l} \left\{1+\frac{\Delta x}{2}S_{0}+\frac{{\left(\Delta x\right)}^{2}}{8}S_{0}^{2}\right\}E_{1}+\left\{{\left(\frac{\Delta x}{\Delta y}\right)}^{2}\left[1+\frac{{\left(\Delta y\right)}^{2}T_{0}^{2}}{8}\right]+\frac{{\left(\Delta x\right)}^{2}}{2\Delta y}T_{0}\right\}E_{2}+\\ \left\{1-\frac{\Delta x}{2}S_{0}+\frac{{\left(\Delta x\right)}^{2}}{8}S_{0}^{2}\right\}E_{3}+\left\{{\left(\frac{\Delta x}{\Delta y}\right)}^{2}\left[1+\frac{{\left(\Delta y\right)}^{2}T_{0}^{2}}{8}\right]-\frac{{\left(\Delta x\right)}^{2}}{2\Delta y}T_{0}\right\}E_{4}+\\ \left\{2+2{\left(\frac{\Delta x}{\Delta y}\right)}^{2}+\frac{{\left(\Delta x\right)}^{2}}{4}\left[S_{0}^{2}+T_{0}^{2}\right]\right\}E_{0}+\Psi_{0}=0. \end{array}$$

Here $S_{0}=R(\psi_{4}-\psi_{2})/2\Delta y$ and $S_{0}=R(\psi_{1}-\psi_{3})/2\Delta x$.

If we use the central difference approximation to the derivatives involved in Poisson equation, Equation (8) at point ‘0’, and then we have
(24)$${\left(\Delta x\right)}^{2}{\left(\Delta y\right)}^{2}E_{0}+{\left(\Delta y\right)}^{2}\left[\psi_{1}+\psi_{3}\right]+{\left(\Delta x\right)}^{2}\left[\psi_{2}+\psi_{4}\right]-2\left[{\left(\Delta x\right)}^{2}+{\left(\Delta y\right)}^{2}\right]\psi_{0}=0.$$

Finally, it is time to address the rest of boundary conditions on vorticity E at the walls. On adopting the method of [51], we can write the required boundary conditions after some simplifications as
(25a)$$E_{b}=\frac{-1}{2{\left(\Delta y\right)}^{2}}\left[-6\psi_{b}+6\psi_{b1}+{\left(\Delta y\right)}^{2}E_{b1}\right]$$
(25b)$$E_{l}=\frac{1}{4{\left(\Delta x\right)}^{2}}\left[-6\psi_{l1}+{\left(\Delta x\right)}^{2}E_{l1}\right]$$
(25c)$$E_{r}=\frac{1}{4{\left(\Delta x\right)}^{2}}\left[-6\psi_{r1}+{\left(\Delta x\right)}^{2}E_{r1}\right]$$
(25d)$$E_{t}=\frac{-1}{2{\left(\Delta y\right)}^{2}}\left[-6\psi_{t}+6\Delta y+6\psi_{t1}+{\left(\Delta y\right)}^{2}E_{t1}\right]$$

The subscripts b, l, r and t denote for bottom, lower, right and top of the cavity respectively, whereas b1, l1, r1 and t1, respectively, signify to the internal grid point most immediate to b, l, r and t.

## 3. Results and Discussion

The boundary value problem (containing Equations (23) and (24) together with boundary conditions (10) and (25)) is then solved numerically by the SOR method with the stopping criteria
(26)$$\max\left|E_{0}^{k+1}-E_{0}^{k}\right|<10^{-5},\;\max\left|\psi_{0}^{k+1}-\psi_{0}^{k}\right|<10^{-5}$$

The selected results are presented in this section. A computer program in the Fortran power station is developed to solve the aforementioned BVP. The numerical outcomes are also tested on different grid sizes, (namely, 62×62, 82×82, and 132×132 ) in order to ensure the grid independence of the solution, which is found satisfactory. However, the results displayed, in the form of streamlines, equivorticity lines and velocity plots, in this section, are based on the finest grid 132×132. In order to validate the computer program and calculations, the results in the form of streamlines are also compared with the existing results of literature as shown in Figure 3. We find that our results are in justification with those in the literature as mentioned in Figure 3. This figure justifies that the present results are in line with those in the literature.

The streamlines for different values of the Stuart number $S_{t}$, the permeability parameter $\alpha_{p}$, the Reynolds number R and the aspect ratio γ are next presented in Figure 4, Figure 5 and Figure 6. Figure 4 is plotted in the absence of magnetic source while Figure 5 is plotted when there exists a magnetic source, i.e., $S_{t}\neq 0$. The streamlines show the development of the primary and secondary vortices.

It can be noted that the number of vortices increases as the Reynolds number intensifies. It means that chaos in the cavity will rise with the applied speed of the lid of the cavity. If $\alpha_{p}=0$ then it means that there is no injection/suction attached. It can be observed that if $\alpha_{p}=0$ then the secondary vortices occur at the right and left corner of the bottom of the cavity. But if $\alpha_{p}\neq 0$ then these vortices will disappear at these positions for fixed value of R, $S_{t}$ and γ. Furthermore, the primary vortex is split into two parts at the bifurcating point (point at which phase of injection and suction is changed). In our case the bifurcating point is x=0.5. Unlike this, if the aspect ratio is increased (i.e., if the depth of the cavity is increased) then the corner secondary vortices join together to form a large sized vortex while eccentricity of the primary vortex increases as shown in Figure 4a,b. Next, the influence of the Stuart number $S_{t}$ is observed in Figure 5.

The nonzero value of the Stuart number indicates the existence of the magnetic source. If the Stuart number increases then the intensity of magnetic field will rise. In this scenario, if $S_{t}\neq 0$ then the right secondary vortex decays in size but the left secondary vortex translates over the bottom towards right wall of the cavity as shown in Figure 4a and Figure 5a. Further, the flow deviates or push towards left as the magnetic source is applied near the right wall for all values of aspect ratios, as shown in Figure 5b,d. Note that Figure 5 and Figure 6 were plotted for moderate Reynolds number and low Stuart number. Figure 6 is presented in contrast to this. It is plotted for low values of Reynolds number but a little bit moderate value of Stuart number. Three primary and two secondary vortices are seen for $S_{t}=0.1$ and R=1 for square cavity if permeability parameter is 0.05 (meaning that $V_{0}=\frac{1}{20}U$) as shown in Figure 6. Overall, if we look at Figure 4, Figure 5 and Figure 6, the common thing is that the primary and secondary vortices will exist for all values of physical parameters: the Stuart number, the Reynolds number, and the aspect ratio and the permeability parameter. However, the size and position of the eyes of the vortices may vary with these physical parameters. Secondly, neither the optimal values of the stream function nor their locations remain the same for all values of physical parameter. They are listed briefly in Table 1.

Figure 7 and Figure 8 are later plotted to understand the behaviors of horizontal speed u and vertical speed v for some chosen values of physical parameters. These figures indicate that if we move from bottom to top both speed components will intensify. The same trend is still observed if we move from left to right of the cavity. Moreover, these velocity components will also rise if the speed of injection and suction enhances. Furthermore, oscillatory motion exists if one of the physical parameters like the Reynolds number, the Stuart number or the permeability parameter increases as shown in Figure 7 and Figure 8.

The equivorticity lines are shown in Figure 9 and Figure 10. These figures designate the growth and decay of vortices in the cavity. Figure 9 shows the effect of permeability while influence of the Stuart number is highlighted in Figure 10. If the bottom of the cavity is not permeable then the uniform vorticity occurs but the nonuniformity or oscillation may be observed if the switch of the permeability is set on as shown in Figure 9. The nonuniformity of the vorticity is also noticed in Figure 10 when the intensity of the magnetic source is increased. The optimal values of the vorticity will help us in designing the strength of the cavity walls. The growth in the maximum value of vorticity will increase the boundary layer thickness that yields the stresses or loading effects on the walls. The optimal values of vorticity are presented in Table 2 as a sample. This table indicates that the location of optimal value of vorticity does not alter with the change in the physical parameters.

Next, Figure 11 is plotted to seek the flow analysis if the suction/injection intervals are reversed opposite to as described in Figure 1. In order to zoom in and highlight the influence of the injection/suction activity, we damped other flow controlling parameters namely the Reynolds number and the Stuart number for a fixed aspect ratio. If we take the reversed intervals of injection and suction, respectively, as 0≤x≤0.5 and 0.5<x≤1.0, it is observed that the streamlines suction/injection as mentioned in Figure 1 are almost mirror image of injection/suction, in parallel to the physical intuition. Furthermore, the size of the secondary vortices reduces if the magnetic intensity intensifies as shown in Figure 11.

Last but not least, the study will be incomplete without the examining the variation of forces due to magnetic fields (which are the Kelvin and the Lorentz forces). Therefore, these magnetic forces are calculated for the problem under consideration. The simplified form of magnitude of the Kelvin force $F_{K}$ and the Lorentz force $F_{L}$ (on using their expressions given after Equation (2)) can be written as
(27)$$F_{K}=\sqrt{{\left(H\frac{\partial H}{\partial x}+K_{K}A_{1}\right)}^{2}+{\left(H\frac{\partial H}{\partial y}+K_{K}A_{2}\right)}^{2}},\;\mathrm{and}\;F_{L}=\sqrt{A_{1}^{2}+A_{2}^{2}},$$
where
(28)$$K_{K}=\chi (1+\chi )K_{1},\;A_{1}=-\left(\frac{\partial \psi }{\partial x}\right)H_{1}H_{2}-\left(\frac{\partial \psi }{\partial y}\right)H_{2}^{2}\;\mathrm{and}\;\;A_{2}=\left(\frac{\partial \psi }{\partial y}\right)H_{1}H_{2}+\left(\frac{\partial \psi }{\partial x}\right)H_{2}^{2}$$

On the behalf of the expressions of magnetic forces given in Equation (27) are examined for different values of flow and geometric parameters. Particularly, for the purpose of presentation here in Figure 12, the chosen values of the parameters are $\left\{K_{K},\;S_{t},\;R,\;\alpha_{p},\gamma \right\}=\left\{0.1,\;0.001,\;10,\;(0,0.5),\;1\right\}.$ It is observed that the Kelvin force decreases as we move from left to right walls of the cavity for all values of the y (i.e., from bottom to top of the cavity) as shown in Figure 12a,c. Unlike this the Lorentz force increases with x as we move from bottom to top of the cavity as shown in Figure 12c,d. Moreover, notice that the influence of magnetic forces rises with the permittivity at the lower wall of the cavity.

## 4. Conclusions

In the current work, we have examined the steady flow of a lid-driven cavity flow of the ferrofluid particles computationally in the presence of variable magnetic source and permeable bottom. The results are presented in form of the streamlines, equivorticity lines, and the variation of the components of speed. A few findings are being summarized as
The number of vortices in the cavity increase with the increase in the Reynolds number.If the aspect ratio is increased (i.e., if the depth of the cavity is increased), then the secondary vortices located at corners of the cavity join together to form a large sized vortex while eccentricity of the primary vortex increases.Neither the optimal values of the stream function nor its location remain the same for all values of physical parameters.The suction and injection create resistance in settlement of solid ferroparticles on the bottom.The location of the optimal values of the vorticity remains constant.If the intervals of the suction/injection are reversed then the streamlines are almost mirror image of each other.The size of the secondary vortices decays if the magnetic intensity is enhanced.The rise in the magnetic forces (the Kelvin and the Lorentz forces) is observed with introduction of the suction/injection chamber.

In this study, we focused on the fluid motion merely, while the topic in hand is immature if the thermal effects are not discussed. Therefore, we shall present them in a future attempt.

## Figures and Tables

**Figure 1 micromachines-10-00373-f001:**
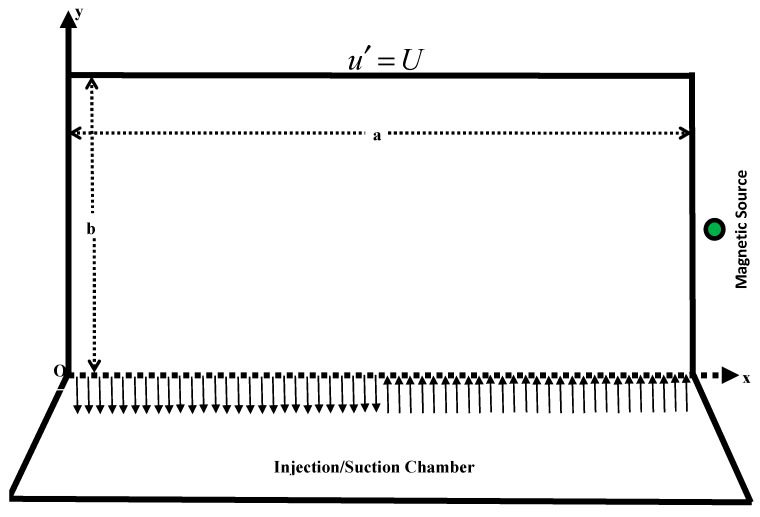
Two-dimensional (cross-sectional) view of the cavity of the problem.

**Figure 2 micromachines-10-00373-f002:**
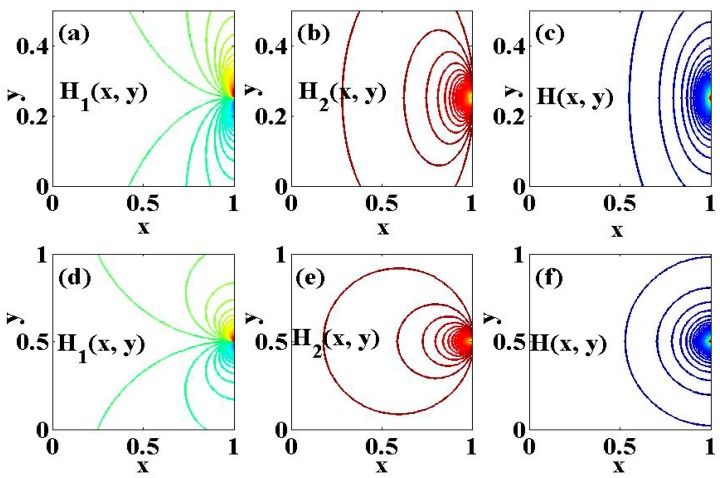
The dimensionless applied magnetic field H and its components for (**a**)–(**c**) γ=0.5 and (**d**)–(**f**) γ=1.

**Figure 3 micromachines-10-00373-f003:**
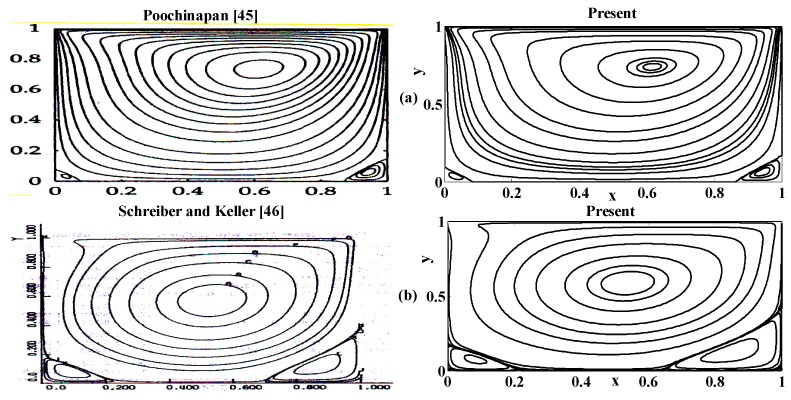
The satisfactory comparison of present results with the literature [45,46] when $\left\{S_{t},\;\alpha_{p},\;\gamma \right\}=\left\{0,\;0,\;1\right\}$ for (**a**) R=100, and (**b**) R=1000.

**Figure 4 micromachines-10-00373-f004:**
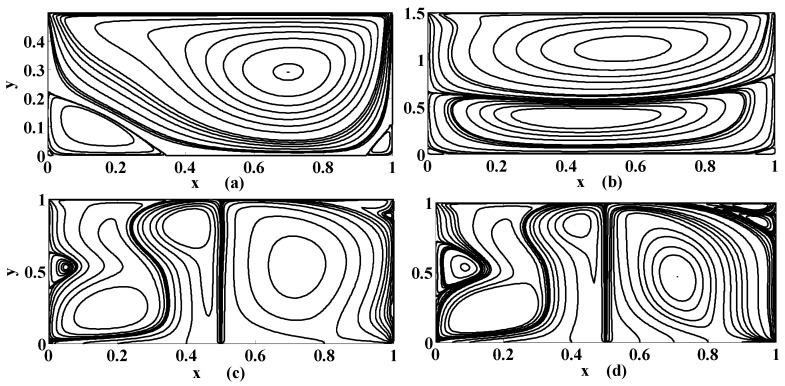
Streamlines for $\left\{S_{t},\;R\right\}=\left\{0,\;400\right\}$ if (**a**) $\left\{\alpha_{p},\;\gamma \right\}=\left\{0,\;0.5\right\}$; (**b**) $\left\{\alpha_{p},\;\gamma \right\}=\left\{0,\;1.5\right\}$; (**c**) $\left\{\alpha_{p},\;\gamma \right\}=\left\{0.5,\;1\right\}$; (**d**) $\left\{\alpha_{p},\;\gamma \right\}=\left\{1,\;1\right\}$.

**Figure 5 micromachines-10-00373-f005:**
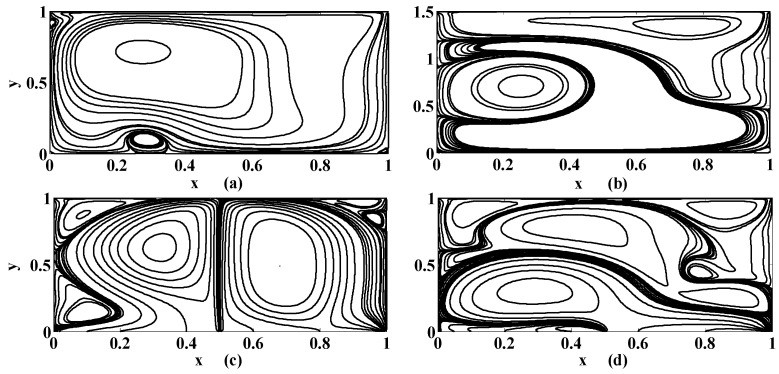
Streamlines for $\left\{S_{t},\;R\right\}=\left\{0.005,\;400\right\}$ if (**a**) $\left\{\alpha_{p},\;\gamma \right\}=\left\{0,\;0.5\right\}$; (**b**) $\left\{\alpha_{p},\;\gamma \right\}=\left\{0,\;1.5\right\}$; (**c**) $\left\{\alpha_{p},\;\gamma \right\}=\left\{0.5,\;1\right\}$; (**d**) $\left\{\alpha_{p},\;\gamma \right\}=\left\{1,\;1\right\}$.

**Figure 6 micromachines-10-00373-f006:**
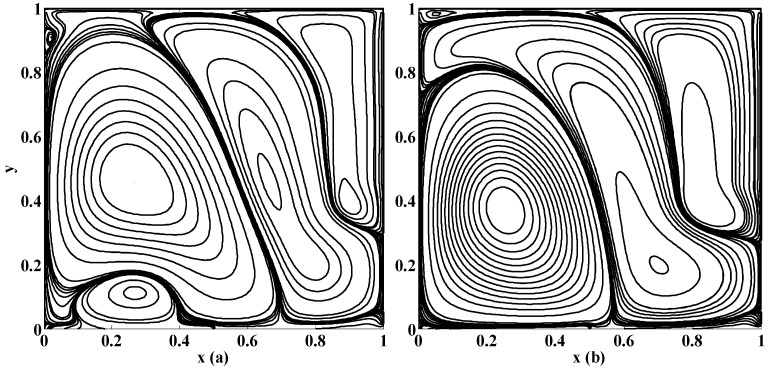
Streamlines for $\left\{S_{t},\;R,\;\gamma \right\}=\left\{0.1,\;1,\;1\right\}$ if (**a**) $\alpha_{p}=0.05$; (**b**) $\alpha_{p}=0.1$.

**Figure 7 micromachines-10-00373-f007:**
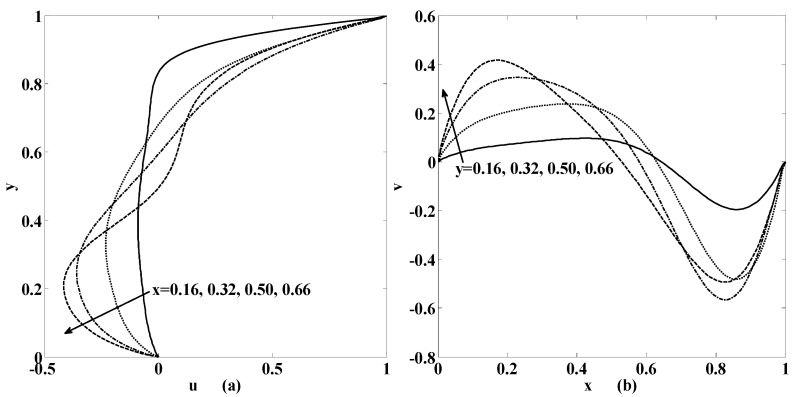
Variation of velocity (**a**) u(x,y); (**b**) v(x,y), at the different positions in the cavity as mentioned within the figure if $\left\{S_{t},\;R,\;\alpha_{p},\gamma \right\}=\left\{0.001,\;10,\;0,\;1\right\}$.

**Figure 8 micromachines-10-00373-f008:**
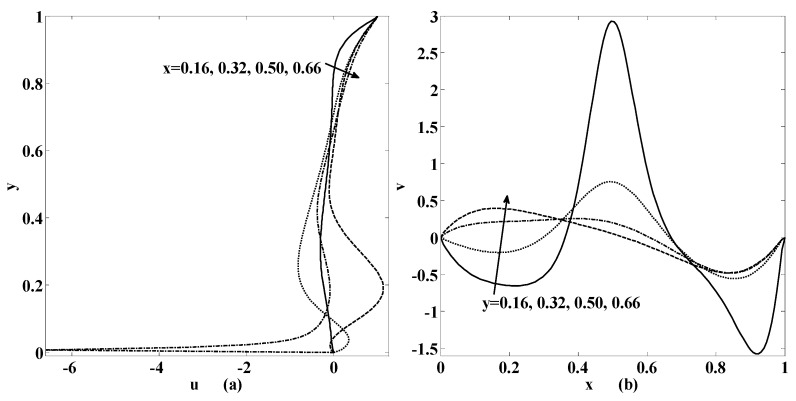
Variation of velocity (**a**) u(x,y); (**b**) v(x,y), at the different positions in the cavity as mentioned within the figure if $\left\{S_{t},\;R,\;\alpha_{p},\gamma \right\}=\left\{0.001,\;10,\;0.5,\;1\right\}$.

**Figure 9 micromachines-10-00373-f009:**
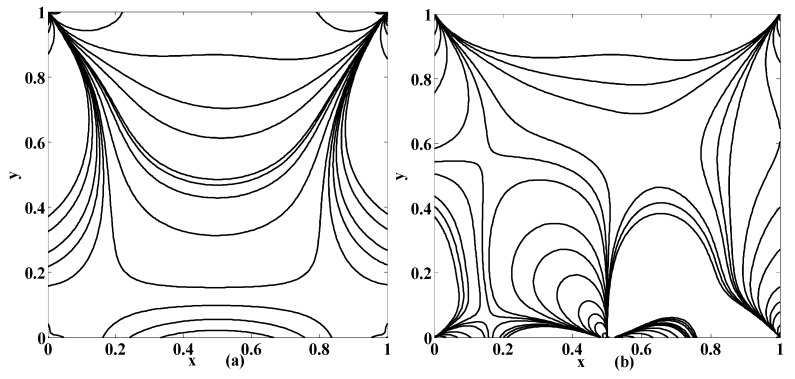
Equivorticity lines if $\left\{S_{t},\;R,\;\gamma \right\}=\left\{0,\;10\;,\;1\right\}$ for (**a**) $\alpha_{p}=0$; (**b**) $\alpha_{p}=0.5$.

**Figure 10 micromachines-10-00373-f010:**
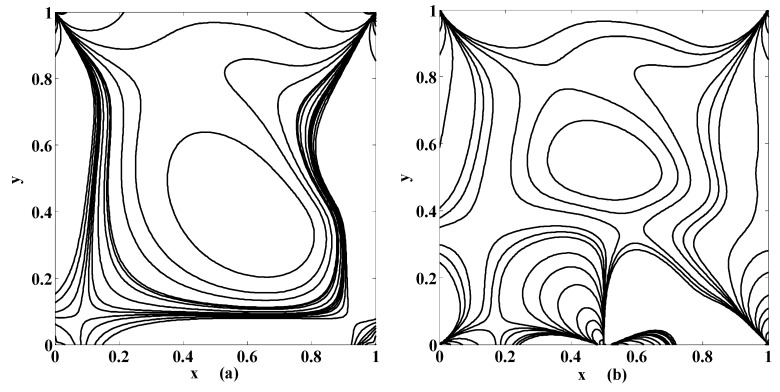
Same as that in Figure 9, except $S_{t}=0.001$.

**Figure 11 micromachines-10-00373-f011:**
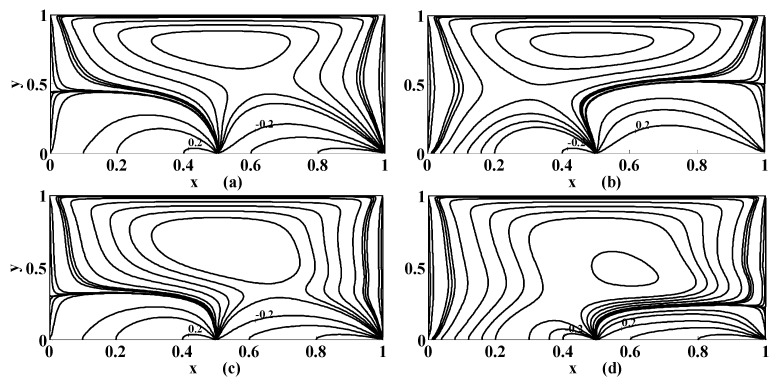
Streamlines with reversed intervals of suction/injection if {R, γ}={0.1, 1} when (**a**) $\left\{S_{t},\;\alpha_{p}\right\}=\left\{0,\;0.5\right\}$; (**b**) $\left\{S_{t},\;\alpha_{p}\right\}=\left\{0,\;-0.5\right\}$; (**c**) $\left\{S_{t},\;\alpha_{p}\right\}=\left\{0.1,\;0.5\right\}$ and (**d**) $\left\{S_{t},\;\alpha_{p}\right\}=\left\{0.1,\;-0.5\right\}$.

**Figure 12 micromachines-10-00373-f012:**
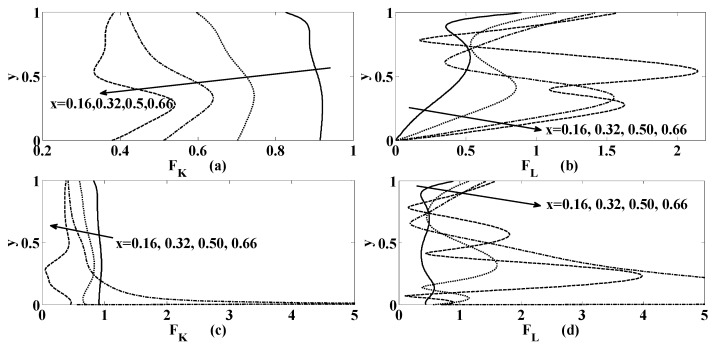
Variation of magnetic forces (**a**), (**c**) the Kelvin force $F_{K}$(**b**), (**d**) the Lorentz force $F_{L}$, at the different positions in the cavity as mentioned within the figure if $\left\{S_{t},\;R,\;\gamma \right\}=\left\{0.001,\;10,\;1\right\}$ for (**a**), (**b**) $\alpha_{p}=0$ (**c**), (**d**) $\alpha_{p}=0.5$.

**Table 1 micromachines-10-00373-t001:** Optimal values of stream function along with its location in the cavity.

Figure	$\psi_{max}$	Location	$\psi_{min}$	Location
Figure 4a	0.0003	(0.137, 0.206)	−0.0813	(0.702, 0.580)
Figure 4b	0.0060	(0.427, 0.389)	−0.1104	(0.557, 1.111)
Figure 4c	0.5881	(0.397, 0.809)	−0.9036	(0.718, 0.542)
Figure 4d	0.9419	(0.412, 0.840)	−1.0917	(0.709, 0.466)
Figure 5a	0.4110	(0.282, 0.107)	−0.4460	(0.267, 0.725)
Figure 5b	0.0583	(0.595, 0.230)	−0.0933	(0.252, 0.710)
Figure 5c	0.7398	(0.321, 0.626)	−0.8774	(0.679, 0.489)
Figure 5d	0.4962	(0.496, 0.000)	−1.0000	(1.000, 0.007)
Figure 6a	1.0443	(0.664, 0.458)	−8.6529	(0.267, 0.458)
Figure 6b	3.0750	(0.252, 0.374)	−15.048	(0.702, 0.198)

**Table 2 micromachines-10-00373-t002:** Optimal values of vorticity along with its location in the cavity.

Figure	$E_{max}$	Location	$E_{min}$	Location
Figure 9a	90.176	(1.000, 0.992)	−109.169	(0.015, 1.000)
Figure 9b	10370	(1.000, 0.007)	−5188.50	(1.000, 0.000)
Figure 10a	90.171	(1.000, 0.992)	−109.185	(0.015, 1.000)
Figure 10b	10377	(1.000, 0.007)	−51887.1	(1.000, 0.000)

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
