# Peer review of "A New Theoretical Approach of Wall Transpiration in the Cavity Flow of the Ferrofluids"

_micromachines, 2019, doi:10.3390/mi10060373_

Round 1
Reviewer 1 Report
The authors of the manuscript describe a theoretical study on how to control vortices in a ferrofluidic system. As the manuscript is written right now it is very hard to understand and clearly see the result. The manuscript must be revised and rewritten (major revision) in a much clearer way. Comments and suggestions on how to change the manuscript are listed below.
Title: the title shall include that the presented work is theoretical. The word "permeability" does that refer to the magnetic permeability (=1+susceptibility) of the magnetic particle system (ferrofluid) in the flow or is it related to the word permeable (suction and injection that used in the text)? Rewrite the title so it more reflects the work.
Abstract: what does "rate of permeability" means? is it some changes in time of the magnetic permeability of the introduced ferrofluid in the flow or is it related to the permeability parameter alfa_p=V_0/U listed in the text? As the text is right now it is hard to understand what the authors would like to say to the reader.
The first sentences in the introduction. Many comments. Some listed below:
- Better to write "Ferrofluids are fluids that contain magnetic nanoparticles in suspension with a particle diameter in the range of .... nm. Please add also a reference.
- Better to write "the magnetic particles are in a single-domain state with a corresponding intrinsic spontaneous magnetization and are typically made of magnetite or maghemite."
- to many "these".
- write "clustering" instead of "clumping".
Introduction page 2:
- when the authors refer to biomedical applications of magnetic liquids, they shall also refer to "Applications of magnetic nanoparticles in biomedicine, Q A Pankhurst, J Connolly, S K Jones and J Dobson, J. Phys. D: Appl. Phys. 36 (2003) R167–R181.
- from the title of reference 12, ref 13 shall be ref 12, or?
- is it really necessary to include so much information of ferrofluids in the introduction, even the chemical reaction is shown when magnetite based ferrofluids are synthesized (see below the comment on the common thread).
- add ”regarding the motion of ferrofluids” when discussing Neuringer and Rosensweigs models.
- more popular to use the Rosensweig model, is this statement really true? Is it not that appropriate models are used for a specific application.
- it is many special parameters (Nusselt number, Stuart number, ...) that is used in the beginning of the manuscript whithout any explanation what they stand for. I think the manuscript could gain very much just to introduce some short explanations of these parameters.
- in what way comes magneto-dielectrics into account here? as well the self-assembly of mono-dispersed particles further down in the text.
- when reading the introduction chapter it is hard to find the "common thread" of all the previous results and references the authors write and list. My suggestion is to rewrite the introduction and to find the "common thread" that is important for the result presented by the author in this manuscript.
Figure texts in general. The figure texts must include some more text that explains what the figure shows in more detail, for instance the different items in figure 1. The figure text in figure 8 cannot just say "same as in figure 7 ...".
Chapter 2:
- is the magnetic source in figure 1 from a permanent magnet? how is the magnetic source positioned/oriented? From where comes the equations (Eq. 1) of the field (H1 and H2)?
- is it possible to decrease the number of mathematical steps and then summarize the used resulting equations?
Chapter 3:
Can the result be summarized/visualized in a clerarer way so it will be more easier for the reader?
Conclusion:
Fill in information regarding Author contributions and Funding.
Author Response
I have attached the file for the reply to reviewer.

Reviewer 2 Report
I find the article interesting. The authors gave a throrough review of the background of the research. I suppose the application of ferrofluids in microfluidics has its potential. However, the content is a bit thin, and unfortunately I have to say the article is poorly written. The typeset of the equations is horrifying. At this stage, the article is not acceptable for publication. Some technical points are given below.
Eq. (1) shows that the magnitude of the magnetic intensity increases with the distance from the source. How is this possible?
I don't see how Eq. (4) follows from Eq. (2). The authors should provide a proper reference or add some more details.
In Eq. (19), the discretisation of S and T is not given.
The results are interesting. However, to understand the physics better I suggest the authors include results for the forces due to the magnetic fields.
In Author Contributions and Funding sections, who are XX and YY?
It seems some of the plots in Figure 3 were copied from other sources without giving proper credit.
Author Response

(The authors gave the same response as above.)

Reviewer 3 Report
The manuscript is in a good shape and the conclusions and results are supported by the analysis presented but also suffer some problems. Therefore, this manuscript can be accepted for publication after some minor revisions as suggested below.
1. In Figure 3, firstly, the font size needs to be increased; secondly, more discussion about the comparison of this work and the literature is required. Authors cannot just say that results shown in figure and let the readers to draw the conclusions by themselves.
2. I would suggest merge Figure 7 and 8 together because the difference between these two figures is just the permeability parameter. Figure 4 and 5 have the same problems. The caption of a figure cannot be like the same as figure xx. If authors have to demonstrate the figures in two separate figures, the caption needs to be re-written.
3. Overall, authors should avoiding using first-person pronouns like "I," "me," "my," "we," "us," etc. In addition, tense of the whole manuscript needs to be consistent.
Author Response

(The authors gave the same response as above.)

Round 2
Reviewer 1 Report
The authors have improved the manuscript by revising the document according to the comments and replied on all comments. Only minor comments listed below.
1. Abstract: good that the authors have included the definition of permeability, but it should also be good if the authors also include in the abstract text what U_0 and V stands for.
2. Introduction: write "fine magnetic particles" instead of "tiny magnetic particles".
Author Response
Subject: Reviewer’s comments and authors’ response for Manuscript ID: micromachines-489422
Dear the Editor-in-Chief,
In the response of the valuable comments raised by the honourable reviewer, the following is the reply to his comments.
AUTHORS’ REPLY TO REVIEWER’ COMMENTS:
Dear Reviewer: I am greatly thankful to you that you guide me and suggest very valuable and highly useful comments. All the corrections/additions are edited in the manuscript as yellow highlighted/green colour font. In addition, I shall feel honour if you give me any further comments/guidance to improve the quality of manuscript.
Sr.No. | Reviewer’s Comments | Reply to Reviewer’s Comment |
1 | The authors have improved the manuscript by revising the document according to the comments and replied on all comments. Only minor comments listed below. 1. Abstract: good that the authors have included the definition of permeability, but it should also be good if the authors also include in the abstract text what U_0 and V stands for.
|
Thank you.
Yes, both the suggestions are great. I comply with your order and put the definition of permeability as well as representation of the U_0 and V in the abstract-lines 8 and 9. |
2 | 2. Introduction: write "fine magnetic particles" instead of "tiny magnetic particles".
| Thank you. It has been replaced as you suggested. |
Regards
Dr. Siddiqui
Reviewer 2 Report
I'm glad to see that the presentation of the artice has been significantly improved. I can recommend it accepted for publication. However, the English still needs improvements. Also, I believe there is a mistake in Eq. 3: I believe the first term on the right hand side is wrong. This must be corrected.
Author Response
Subject: Reviewer’s comments and authors’ response for Manuscript ID: micromachines-489422-Round 2
Dear the Editor-in-Chief,
In the response of the valuable comments raised by the honourable reviewer, the following is the reply to his comments.
AUTHORS’ REPLY TO REVIEWER’s COMMENTS:
Dear Reviewer: I am greatly thankful to you once again that you guide me and suggest very valuable and highly useful comments.
Sr.No. | Reviewer’s Comments | Reply to Reviewer’s Comment |
1 | I'm glad to see that the presentation of the article has been significantly improved. I can recommend it accepted for publication. However, the English still needs improvements. Also, I believe there is a mistake in Eq. 3: I believe the first term on the right hand side is wrong. This must be corrected.
| Thank you for acknowledging my efforts and accepting the article for publication. Yes the English still needs improvement. It will be further improved during its final proof reading that was made by my coauthor. He is very learned person. I think that Eq. 3 is correct. I have rechecked my calculation again and again. I have also verified with the literature for example kindly see Reference [28] page 077103-2. This paper is also attached along with letter or will be sent to editor. |
Regards
Dr. Siddiqui
